# Shortwave Irradiance (1950 to 2020): Dimming, Brightening, and Urban Effects in Central Arizona?

**Anthony Brazel** [1,*] **and Roger Tomalty** [2]

1   Urban Climate Research Center, School of Geographical Sciences & Urban Planning, Arizona State University, Tempe, AZ 85287-5302, USA

2   Independent Researcher, 10301 N. 70th St. #12, Scottsdale, AZ 85253, USA; rtomalty@gmail.com

*   Correspondence: abrazel@asu.edu

**Abstract:** The objective of this study was to evaluate long-term change in shortwave irradiance in central Arizona (1950–2020) and to detect apparent dimming/brightening trends that may relate to many other global studies. Global Energy Budget Archives (GEBA) monthly data were accessed for the available years 1950–1994 for Phoenix, Arizona and other selected sites in the Southwest desert. Monthly data of the database called gridMET were accessed, a 4-km gridded climate data based on NLDAS-2 and available for the years 1979–2020. Three Agricultural Meteorological Network (AZMET) automated weather stations in central Arizona have observed hourly shortwave irradiance over the period 1987–present. Two of the rural AZMET sites are located north and south of the Phoenix Metropolitan Area, and another site is in the center of the city of Phoenix. Using a combination of GEBA, gridMET, and AZMET data, annual time series demonstrate dimming up to late 1970s, early 1980s of −30 W/m$^2$ (−13%), with brightening changes in the gridMET data post-1980 of +9 W/m$^2$ (+4.6%). An urban site of the AZMET network showed significant reductions post-1987 up to 2020 of −9 W/m$^2$ (3.8%) with no significant change at the two rural sites.

**Keywords:** shortwave irradiance; Phoenix; Arizona; GEBA; gridMET; AZMET; dimming and brightening; trends; urbanization; PM10

## 1. Introduction

In the 20th and 21st centuries, heightened interest has revolved around the phenomenon of an apparent global and regional dimming and brightening of the Earth as shortwave irradiance (K↓) has undergone decadal timescale changes [1,2]. Evaluation of measurements and modeling of Earth's downward shortwave irradiance (and other fluxes) has become essential to support the studies of Earth's energy and water balance for the expediency of understanding global and regional climate change [3,4]. In [3], the authors describe the Global Energy Balance Archive (GEBA), a database of worldwide energy fluxes of the Earth's surface including shortwave irradiance data. Over 2500 stations are entered in this database. Several studies using GEBA records have pointed to declines in shortwave irradiance at many sites in Europe, the Baltic, South Pole, Germany, and Russia between the 1950s and 1980s as pointed out in the review article by [2]. The reduction phenomenon has been labeled as "global dimming" [5]. Recent studies have shown a trend reversal and recovery since the 1980s, which has been labeled "brightening" [6]. There remains much uncertainty in these trends and their causes for given locations on Earth, as changes in natural and anthropogenic aerosols, and cloudiness both play roles in impacting trends [2]. Dimming phases typically show reductions of −3 to −9 W/m$^2$, while brightening phases range from 1 to 4 W/m$^2$; and as stipulated in the review of [2], it is more likely that recovery values are in the lower bound of this range due to urbanization effects. The study of [7] demonstrates that the sites in the United States have shown large dimming reductions estimated at −19 W/m$^2$ or −10% over the period 1961–1990. Clear sky declines over this same period are cited as −8 W/m$^2$. In scanning values cited in [1] for comparable





arid or semi-arid sites to the environment in Phoenix, Arizona, values of dimming have been calculated for Israel ($-9$ W/m$^2$ or $-5\%$ for 1954–94), Egypt ($-13$ W/m$^2$ or $-6\%$ for 1968–94), and an extreme example from Israel of some $-58$ W/m$^2$ over the period $1958-1985$ (attributed to severe pollution effects close to the station). A study by [8] for the brightening period of 1995–2007 for a number of sites in the continental USA resulted in a $+8$ W/m$^2$ or $+4.4\%$ increase in shortwave irradiance. In sum, global change scientists studying dimming and brightening of the Earth have been confronted with the complexities of sorting out effects of pollution locally vs globally in assessing shortwave irradiance variations over space and time. As [2] has pointed out, much uncertainty persists on the recovery or brightening phase.

In the fields of urban climatology, agricultural meteorology, satellite technology and communication, and allied fields, large strides have been made in regional and mesoscale monitoring, deployment of special network data collection across local to global scales, and using satellite technology and modeling to estimate shortwave irradiance, especially since the 1970s–80s [9]. Database archives and reports have been assembled for urban, regional, and global scales to address changes over time in the Earth's energy and water budget [3,4]. Researchers have previously studied shortwave irradiance as part of urban energy balance studies for a host of cities around the world with highly variable urban vs. rural results, which further complicates attempts at the generalization of impacts of urbanization on this fundamental input of radiation [10]. The range of differences between rural and urban radiation values among cities may be 0 to $+33\%$ due to varying urban boundary layer differences, geographic location, pollutants, and/or the result of methodologies and time frames utilized. With better monitoring since the 1970s–80s and increases in special networks near and in urban areas, the brightening phase may be more accurately depicted.

The objective of this paper was to explore the changes in shortwave irradiance using three databases each covering portions of the years 1950–2020 for the location of central Arizona in the Southwest United States, an area of desert terrain occupied by one of the largest desert cites in the world—Phoenix and its metropolitan area. The analysis below tests the degree to which dimming and brightening are evident in these databases, which each cover portions of this 70 year period. The sites and data used for this study are discussed in the Methods section and two separate analyses are presented: (a) annual time series of shortwave irradiance to display the degree to which dimming and brightening patterns are evident in solar records, and (b) the possible effects of pollution impacting the radiation recorded in central Arizona and the Phoenix area. To our knowledge, this is a first study to interpret this 70 year period for the Southwest desert region and specifically central Arizona. In the Methods section, we discuss the study area, databases used, issues of data quality, time series created for analysis, and statistical procedures to determine the existence of dimming and brightening and the degree of confidence in the findings. We present a Results section that analyzes a defined dimming and brightening period, and analysis of a recent year (2019) to explore urban and rural noon time solar transmissivity differences for clear days, an indicator of the urban pollution impact on the solar record for this area. This is followed by a Conclusions section that provides estimates of overall changes from 1950–2020 and a discussion of our findings with other literature.

## 2. Methods

### 2.1. Study Area and Sites Used

The study area is shown in Figure 1 and information on the sites and databases are listed in Table 1 and explained below. The PM10 records are used in a later section and the AZMET (an automated Agricultural Meteorological Network [11]) and National Weather Service's Phoenix Sky Harbor Airport sites are discussed together with other records in later sections.

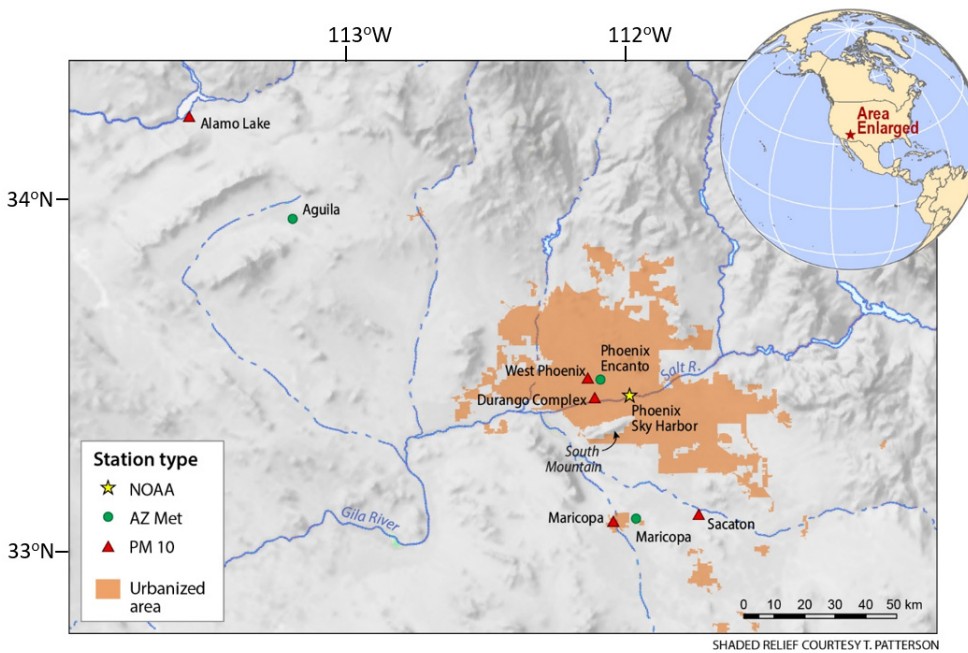

**Figure 1.** Study area of Phoenix, AZ in the desert SW USA. Sites used are shown with symbols and legend refers to the type of site, all mentioned in the paper. The orange pattern shows the urbanized area. Upper left corner of map = 34.38°N, 113.93°W; lower right = 32.76°N, 110.95°W.

**Table 1.** Sites used in the study. See Figure 1 for the locations.

| Stations | Source | Lat°N | Long°W | Elevation Meters | Solar/PM10 Instruments | Land Cover |
|---|---|---|---|---|---|---|
| Aguila | AZMET [1] | 33.95 | 113.19 | 657 | LiCor LI200 Silicon Cell Pyranometer 400 to 1100 nm | Rural agricultural field |
| Phoenix Encanto | AZMET | 33.48 | 112.10 | 334 | LiCor LI200 Silicon Cell Pyranometer 400 to 1100 nm | Suburban golf course |
| Maricopa | AZMET | 33.07 | 111.97 | 362 | LiCor LI200 Silicon Cell Pyranometer 400 to 1100 nm | Rural agricultural field |
| Phoenix Sky Harbor | GEBA [2] | 33.43 | 112.0 | 340 | Eppley model 50 Pyrheliometer 1951-75; after 1975 Eppley Pyranometer PSP 285 to 2800nm | Urban airport |
| Alamo Lake PM10 | *Air Now* [3] | 34.24 | 113.56 | 398 | SSI High Volume Samplers | desert |
| West Phx PM10 | *Air Now* | 33.48 | 112.14 | 334 | SSI High Volume Samplers | suburban |
| Durango Complex PM 10 | *Air Now* | 33.43 | 112.12 | 331 | SSI High Volume Samplers | urban |
| Maricopa PM 10 | *Air Now* | 33.06 | 112.05 | 358 | SSI High Volume Samplers | rural |
| Sacaton PM10 | *Air Now* | 33.08 | 111.75 | 393 | SSI High Volume Samplers | Rural |

[1] AZMET sites from [11]. [2] GEBA is an archive site described in [3] and requires permission. [3] *Air Now* is website described in [12].

The Sky Harbor International Airport site is located near a dry bed of the Salt River within the urbanized area of the city of Phoenix and is subjected to high air pollution episodes [13]. The AZMET site of Phoenix Encanto, simply called Encanto in this paper, is situated in the middle of the city of Phoenix, not far from Sky Harbor International Airport and in a dense residential area on the western edge of a golf course on year round irrigated

turf. This site is also not far from an air quality site used in this study (West Phoenix PM10) and was chosen as a shortwave irradiance urban site. North and south of Phoenix are rural AZMET sites on entirely agricultural landscapes; one northwest of Phoenix at the small town of Aguila, and one near the town of Maricopa ~50 km to the south (Figure 1). Aguila is higher in elevation and out of the air shed of the Salt River Valley. We were less confident in using this site, since it is over 100 km from the urban area and 323 m higher in elevation, but is out of the Salt River Valley airshed. PM10 data were from the nearby monitoring sites of Sacaton and Maricopa near the Maricopa AZMET site, and the Alamo Lake PM10 monitoring site is representative of background rural values and is north of the Phoenix area near Aguila (note all sites in Figure 1). Three databases are used in this study and are reviewed in Section 2.2.

*2.2. Data Bases Used and Methods Employed*

2.2.1. The GEBA Dataset

GEBA is a Global Energy Balance Archive database of worldwide scope and consists of monthly and annual data from measurements of energy fluxes at the surface of the Earth. It is maintained by ETH Zurich in Switzerland [3]. We were granted access to data and downloaded several records from the Southwest desert in the United States. The Phoenix Sky Harbor International Airport solar records, which span the period 1950 to 1994, were mostly complete. We surveyed other data from the archive for the Southwest desert region of Arizona, southern California, and southern Nevada. The data for Las Vegas, Nevada; El Centro, California; and Tucson, Arizona (not shown on Figure 1) were available but not of sufficient completeness and/or length of time interval to compare with the more complete records of the Phoenix data. For Phoenix, the coverage 1950 to 1978 (336 months) only had 4% missing values. These months were interpolated to produce an annual total by using averages prior to and after missing months. We tested this simple approach by voiding a given month, estimating it, and comparing it to actual data for that month. This produced an error in annual totals of $\pm 1 \, \text{W/m}^2$ for five of the years (1951, 1958, 1970, 1973, and 1977). Very few months appeared in the archive from 1979 to 1994. Out of 180 months in this period, only 24 months were available. Details of these data are discussed below.

2.2.2. The gridMET Dataset

The gridMET database is a dataset of daily high-spatial resolution (~4-km, 1/24th degree) surface meteorological data covering the contiguous U.S. from 1979 to the present [14]. The gridMET database puts together spatial attributes of gridded climate data from PRISM with temporal attributes (and additional variables) from regional reanalysis (NLDAS-2) using climatically aided interpolation. The result is a spatially and temporally gridded dataset of surface meteorological variables including shortwave irradiance. Validation of the resulting gridded surface meteorological data was conducted against an extensive network of weather stations. Details of the database are discussed in [14,15]. It is noted in [15] that the gridMET data will likely not capture microclimates that arise at spatial scales finer than the resolution of the grid (<4-km). The gridMET solar radiation is interpolated from NARR/NLDAS-2, which has a 32-km spatial resolution. Solar radiation from gridMET is not adjusted for topographic effects, but instead is provided for a planar surface. Some studies have shown that NLDAS2 downward shortwave radiation shows a positive bias over much of North America. The NARR downward shortwave radiation field in the NLDAS-2 forcing files ("A" files) is bias-corrected to the University of Maryland Surface Radiation Budget (SRB) dataset produced under the auspices of the GEWEX Continental Scale International Project (GCIP) and GEWEX Americas Prediction Project (GAPP) [4]. Data from the GOES-8 satellite were processed to produce hourly estimates of downward shortwave radiation fluxes. A ratio-based bias correction to the reanalysis downward shortwave radiation field has been completed [16].

For our study, we used the climate tool system on the website of [15] to download monthly and annual K↓ data, which are available (1979–2020) for a 4-km grid centered

on any specific location. We additionally used the useful climate tool interface of [16] to obtain annual trend information provided on the website (statistics include trend per decade in W/m$^2$, r value, and significant level) for 4 km grid values centered on a grid box for the whole Southwest desert region, and 4 km grids centered on Las Vegas, Tucson, El Centro (which all do have some records in GEBA), and our AZMET site locations of Aguila, Encanto, and Maricopa.

The monthly records of the Phoenix data from GEBA overlap in time with gridMET data from 1979–1994. The sample size was small (N = 24 months); however, we decided to compare GEBA with gridMET to see whether the data were at least somewhat similar from month to month (see Tables 2 and 3). The findings showed higher gridMET values relative to GEBA. This makes sense, given the provisos of the gridMET database, as it is likely that gridMET does not capture local effects such as air pollution and details of topographic influences and microclimates [15], wherein the Phoenix station would particularly be under these influences. The regression results indicate highly significant r value, small bias, standard errors of coefficients, and similar 95% confidence intervals using bootstrapping with N = 1000 samples.

**Table 2.** Descriptive statistics of overlap months (1979–94) of GEBA and gridMET K↓ (W/m$^2$).

| Dataset | N (Months) | Mean W/m$^2$ | Std Dev | Bias | Std Error | Lower 95% CI | Upper 95% CI |
|---|---|---|---|---|---|---|---|
| GEBA | 24 | 191.4 | 72.49 | −0.001 | 14.56 | 164.1 | 220.8 |
| gridMET | 24 | 195.4 | 73.34 | −0.067 | 14.74 | 169.1 | 225.6 |

**Table 3.** Linear regression gridMET vs. GEBA. Bootstrapping N = 1000. GEBA = A*(gridMET) + B.

| Regression | N (Months) | Pearson's r | Sig. | A | B | A,B Bias | A,B Std Error | Lower 95% CI A,B | Upper 95% CI A,B |
|---|---|---|---|---|---|---|---|---|---|
| gridMET vs. GEBA | 24 | 0.99 | 0.00 | 1.0 | 0.09 | 0.03, −0.17 | 0.039 6.95 | 0.915, −15.42 | 1.073, 12.91 |

### 2.2.3. The AZMET Dataset and Sites Used in Study

AZMET instruments, sites, land cover, and quality control procedures are presented in documents found on the AZMET website [11]. All monthly solar data used were downloaded from [11]. Of importance to this paper are the solar sensors, location relative to urban effects, and completeness of data during the period 1987–2020. Instruments used were LiCor200 pyranometers with wavelength response of 400 to 1100 nm. It should be noted that this spectral range differs from Eppley or other radiometers designed to measure K↓. However, visible ranges of the solar spectrum do overlap. In addition to other variables in this agricultural network, only global shortwave irradiance is monitored at these sites and not separate diffuse or direct beam components. AZMET personnel replace pyranometers every 12 months. If calibration shows significant problems, instruments are refurbished and returned to a site. Although the network consists of over 20 sites in central and southern Arizona, only three stations were chosen—Aguila (rural), Encanto (urban), and Maricopa (rural)—because they have a long and complete record starting from 1987 (Figure 1) and are near and within the Phoenix Metropolitan Area. The entire AZMET network as of 1996 was previously used for the months of January to November of 1996 to aid in verification of resultant NLDAS-2 estimates of downward K↓ calculations underlying the gridMET database [4]. The monthly mean regression results shown in [4] list an r value of 0.99, root mean square error of 13.92 W/m$^2$, bias of −8.37 W/m$^2$, with N station months of 231, for AZMET vs. data labeled as GCIP/Sat (precursor to gridMET data).

2.2.4. Clear Day Analysis for Sample Year 2019: Transmissivity and PM10 Data

We suspected that the urban site of Encanto has effects from local pollution in the Salt River Valley. As a result, we chose a recent year (2019) as an example to more closely look at this site and a nearby rural site (Maricopa). A selection of clear days was determined from the analysis of all days in 2019, the most recent year before the pandemic years. We believe that 2019 is more typical of industrial and population activity than during the recent pandemic. We accessed cloud cover data available from Sky Harbor's International Airport records [17]. A ceilometer was used to detect multiple cloud layers above the station and coverage was additionally assigned and is listed in [17]. Data for clear skies and various cloud types on a 5 minute basis were obtained from the station database. We analyzed all daylight hours and only accepted mid-day periods with designated totally clear skies for every 5 minute period for a two hour block on either side of the noon hour. In addition, for each day chosen in this way, hourly solar values were plotted on the same plot for the stations Encanto and Maricopa to further ensure only clear days were selected across the two sites, since Maricopa is some 50 km south of the Phoenix area. Once we selected these days, we also reviewed data from [17] for the Tucson Airport to the south to ensure that clear sky data were coincident with the Phoenix record. This process resulted in the selection of 39 days spread throughout all months in 2019. The PM10 data for sites listed in Table 1 were accessed from the EPA's *Air Now* website [12] for the 39 days and are only daily values. Thus, for this paper, only day to day values were available, since hourly PM10 data were not readily available to match with the hourly solar data. We calculated transmissivity values for noon Local Standard Time (LST) from solar data in the AZMET archives for Encanto and Maricopa, as the database is not based on solar noon times. Resultant calculations may be slight underestimates of transmissivity as solar noon occurs ~30 minutes after time zone LST.

## 3. Results

### 3.1. A Dimming Period 1950–1978

For the dimming period, we first discuss issues of data measurements and uncertainty. Direct personal access of data obtained from personnel of the Phoenix National Weather Service Office (NWS) of the 1950 to 1970 solar data was accomplished by [18] in order to study atmospheric transmissivity and air pollution for clear days during this 21 year period. Relative to instruments utilized at the station, [18] reported that Eppley pyranometers were in use over this time period, and it was thought that progressive deterioration of the sensitivities of the instruments took place. It is cited in [18] that it was necessary to proportionally adjust data over the period of each instrument change (not defined in [18]) so that the results at the end of each period matched smoothly with the initial results of each newly installed pyranometer (it should be noted that for part of this time, instrumentation was an Eppley Pyrheliometer, Table 1). How many adjustments were made is not reported by [18], nor do we have access to the adjustment information for this total period. The study of [19] helps to retrospectively shed some light on the instruments and data from the dimming period for Phoenix and other stations. The study documented apparent changes required as a consequence of calibrations of instruments at several sites across the western United States including the Phoenix station. The concerns at the time were issues of instrument errors, leading to misuse of data for many applications in the solar energy industry and in interpreting national solar radiation maps [20]. We show selected results from [19] in Table 4. First, note the variations of calibration corrections needed across the sites listed. Values range from −1.0%/yr to +7.8%/yr. The desert Inyokern station is in an arid rain shadow on the east side of the southern Sierra Nevada Mountains in southern California. The errors detected of +16.7% prior to 1966 cast doubt of that location supposedly having the highest solar radiation received in the United States [19,20]. Further investigation at many other national sites by NWS across the country showed corrections needed ranging from +0.8% to +15.0%. The data from the 1966 NWS calibration check for Phoenix notes that a new instrument has been in use since 1962, and that over the four

year period up to 1966, a calibration correction of +2.9% was needed. Errors have been extensively analyzed by [21] and indicate errors from monthly to annual means of some 3–5% for some GEBA data. Exact errors at specific sites would best be documented in detail by those who have original records of the history of the instruments. It should be noted that the known errors for the Phoenix station are similar to those reported by [21] from the GEBA database analysis.

**Table 4.** Selected USA sites and calibration of solar instruments (data from [19]).

| Station | In Use Since | Date of Field Comparison | Correction (%) Total | Correction (%) Mean Annual |
|---|---|---|---|---|
| Phoenix, AZ | June 1962 | Sept 1966 | +12.2 | +2.9 |
| Fresno, CA | Feb 1963 | Sept 1966 | +4.4 | +1.3 |
| Inyokern, CA | Nov 1950 | Sept 1966 | −16.7 | −1.0 |
| Ely, NV | March 1963 | Sept 1966 | +6.9 | +2.0 |
| Davis, CA | July 1965 | Sept 1966 | +7.8 | +7.8 |
| Albuquerque, NM | Unknown in 1966 | Sept 1966 | +2.8 | +2.8 |
| Las Vegas, NV | Nov 1965 | Sept 1966 | +0.8 | +0.8 |

From [19], the Phoenix correction needed for the annual data at least for the period 1962–1966 is +2.9% or ~6 W/m$^2$. We assigned this correction and calculated the descriptive statistics, ran linear regression with bootstrapping, and created a corrected and uncorrected time series plot (Figure 2). The plot did not show a large drop in the period 1963–1967, even with the known correction. Although we did not analyze cloudiness in this paper, the summer months during this time were unusually cloudy. We know this by accessing a synoptic classification for Phoenix that spanned the period 1948 to the present, which includes daily frequencies of air mass types [22]. These years included increased days of Maritime Tropical air masses that likely account for the lower solar receipt during that time. There may be other possibilities to explain this large drop.

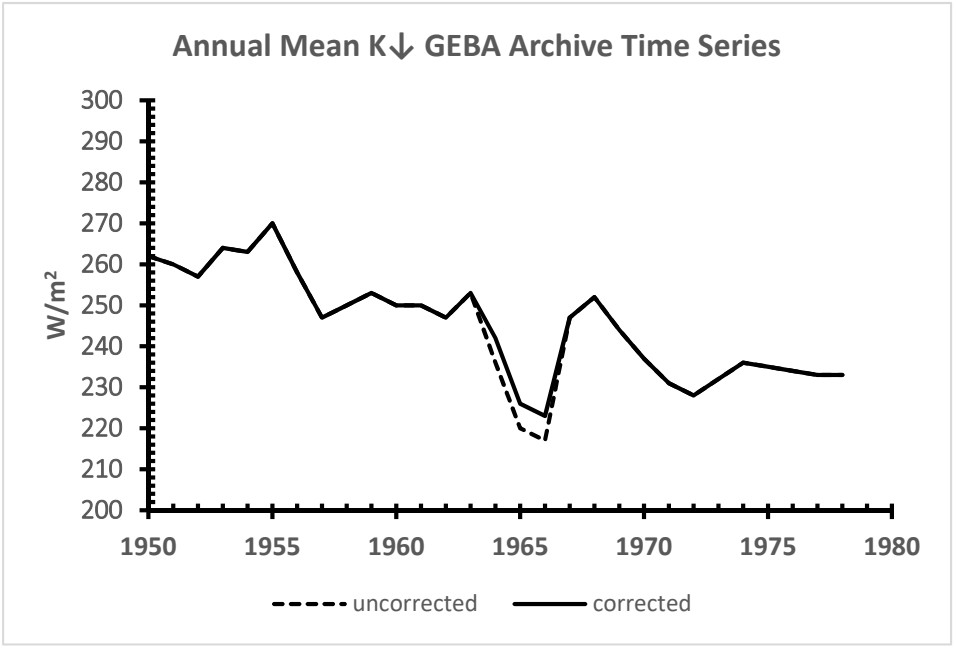

**Figure 2.** Annual mean shortwave irradiance K↓. Solid line-data from GEBA with interpolated missing years and correcting for known errors in instruments. Dashed line represents raw data from the GEBA Archives for the correction period.

We extracted data from Figure 2 shown in [18]. Unfortunately, we do not have the original digital data used by [18]. In [18], annual average atmospheric transmittance values for all clear days during the entire 1950–70 period (% possible sunshine read 100% and sky cover was zero for the day) were plotted on a three year average basis. We converted the transmittance data in [18] to K↓ values in W/m$^2$ by using a solar calculator to obtain extraterrestrial radiation [23]. The transmittance data times these values yielded the estimates of K↓. We then calculated the GEBA records on the same three year basis used in [18], and plotted both sets of data in Figure 3.

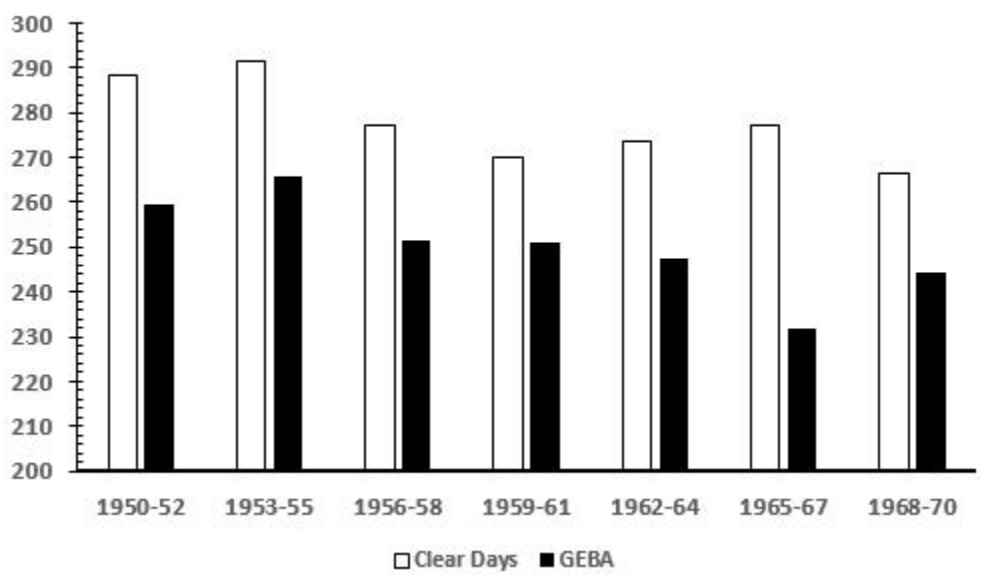

**Figure 3.** Three-year mean annual K↓ (W/m$^2$). Clear days extracted from Figure 2 in [18] and GEBA data for all days.

Understandably, the clear day values were higher than the GEBA records because the GEBA data include all days. A downward change from 1950–52 to 1968–70 for clear days from [18] is evident. The resultant decrease for the clear day values was −7.0% [18]. The period 1965–67 for the GEBA data departs significantly from other periods and is also shown in Figure 5. However, the value for the clear day mean for this period did not drop because of any apparent instrument error, indicating that cloudiness, as suggested above, significantly impacted all the sky annual totals shown in the GEBA record. Since we added a correction that we know about for this same period (possible instrument errors), it is likely that the remaining dip in the record for 1965–67 is related more to synoptic effects [22].

Tables 5 and 6 show that there is a significant dimming (reduction) for both uncorrected and corrected time series over the 29 year period with high negative r values, significant at $p = 0.000$. The biases, standard errors, and lower and upper confidence levels of slope A and intercept B of the linear equation are shown to express the degree of uncertainty in the relationships for the uncorrected and corrected time series regression. For the corrected series, slope A bias was 0.8%; standard error was 9%; lower and upper 95% CI were −1.52 to −1.03. The bias of B intercept was 0.7%; standard error, 9%, with 95% CI from 2274 to 3237. The mean for the dimming period was 245.4 with standard deviation of ±12.8 W/m$^2$.

**Table 5.** Annual shortwave irradiance descriptive statistics of dimming period 1950–1978 with bootstrapping (N = 1000). For the GEBA corrected time series, the lower and upper 95% confidence intervals were 240.8 and 250.2. Only 4% missing months. These were interpolated for an annual total by using averages prior to and after missing months. This produced an error in annual totals of $\pm 1$ W/m$^2$ for five of the years (1951, 1958, 1970, 1973, and 1977).

| Dataset | N (years) | Mean W/m$^2$ | Std Dev | Bias | Std Error | Lower 95% CI | Upper 95% CI |
|---|---|---|---|---|---|---|---|
| GEBA Uncorrected | 29 | 244.8 | 13.62 | 0.130 | 2.33 | 240.3 | 249.7 |
| GEBA Corrected | 29 | 245.4 | 12.8 | 0.025 | 2.32 | 240.8 | 250.2 |

**Table 6.** Linear regression of annual GEBA data with bootstrapping (N = 1000).

| Dataset | N (years) | Pearson's r | Sig. | A | B | A,B Bias | A,B Std Error |
|---|---|---|---|---|---|---|---|
| GEBA uncorrected | 29 | −0.78 | 0.000 | −1.29 | 2695 | −−0.003, 5.166 | 0.13, 255.1 |
| GEBA corrected | 29 | −0.83 | 0.000 | −1.24 | 2679 | -0.010, 20.1 | 0.12, 242.1 |

K↓(in W/m2) = A*(year) + B. For the GEBA corrected series, 95% lower and upper confidence intervals for A were −1.52 and −1.03; for B, 2274 and 3237.

If we solve the regression equation using A and B values for the GEBA corrected series for the start and end points, the dimming will result in some ~30 W/m$^2$ from 1950–78 or −13%. The study of [7] indicates a clear sky reduction for the United States for 1961–90 of −8 W/m$^2$, almost the same for the Phoenix area as determined by [18] for the period 1950–70. The all sky value in [7] as a whole was −19 W/m$^2$ (1961–90), but this value includes sites that may not have pollution effects, unlike Phoenix.

### 3.2. A Brightening 1979–2020

For Figure 4, we placed the GEBA (1950–78) time series on a plot with the subsequent time series of gridMET (1979–2020); Aguila, Encanto, and Maricopa (1987–2020). Notable features are more varied year to year for the surface stations compared to the smoother gridMET data, and an increasing trend (brightening) evident for the gridMET time series.

For the period 1979–2020, we calculated similar descriptive and linear regression statistics as we did for the dimming period. Tables 7 and 8 list the results.

**Table 7.** Descriptive statistics of annual data with bootstrapping (N = 1000). * CI = confidence interval; gridMET[a] is 1979–2020; gridMET[b] is 1987–2020; Aguila is 1987–2020; Encanto is 1987–2020; Maricopa is 1987–2020.

| Dataset | N (yrs) | Mean (W/m$^2$) | Std Dev | Bias | Std Error | Lower 95% CI * | Upper 95% CI * |
|---|---|---|---|---|---|---|---|
| gridMET[a] | 41 | 235.9 | 4.48 | 0.005 | 0.68 | 234.6 | 237.2 |
| gridMET[b] | 34 | 237.1 | 3.82 | 0.019 | 0.62 | 235.8 | 238.3 |
| Aguila | 34 | 237.0 | 5.91 | 0.021 | 0.99 | 234.9 | 238.9 |
| Encanto | 34 | 231.5 | 6.62 | −0.045 | 1.15 | 229.1 | 233.6 |
| Maricopa | 34 | 239.6 | 7.23 | −0.037 | 1.25 | 237.0 | 242.1 |

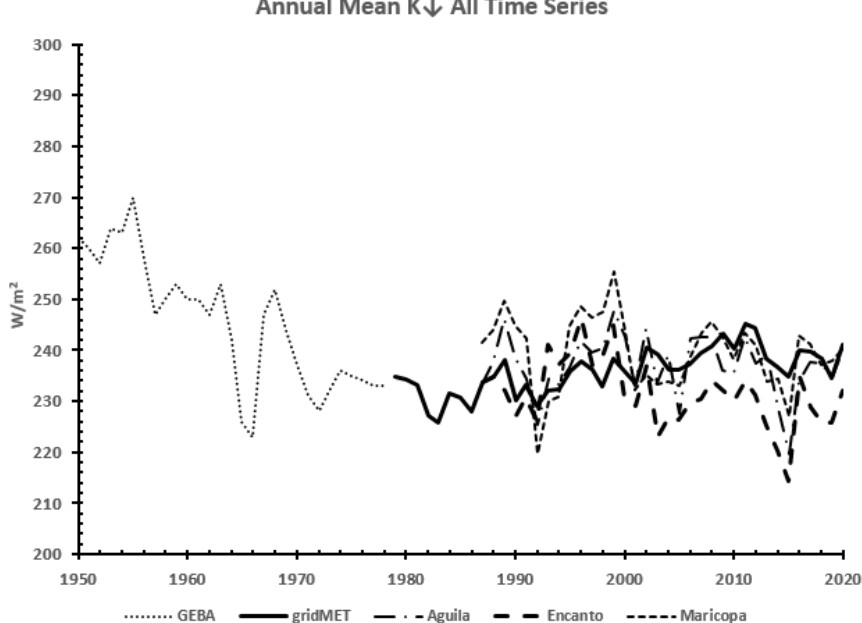

**Figure 4.** Annual trend GEBA, gridMET; Aguila, Encanto, Maricopa.

**Table 8.** Linear regression K↓$_{ANNUAL}$ = A*(year) + B with bootstrapping (N=1000). Time series defined in Table 7.

| Dataset | N (yrs) | Pearson's r | Sig. l | A | B | A,B Bias | A,B Std Error | A,B Lower 95% CI | A,B Upper 95% CI |
|---|---|---|---|---|---|---|---|---|---|
| gridMET[a] | 41 | 0.70 | 0.00 | 0.26 | −276 | 0.000, −0.5 | 0.045, 90 | 0.18, −468 | 0.35, −116 |
| gridMET[b] | 34 | 0.60 | 0.00 | 0.23 | −223 | 0.003, −5.0 | 0.056, 113 | 0.12, −461 | 0.35, −14 |
| Aguila | 34 | −0.11 | 0.27 | −0.07 | 366 | −0.005, 9.7 | 0.104, 207 | −0.28, −30 | 0.13, 801 |
| Encanto | 34 | −0.44 | 0.01 | −0.29 | 816 | −0.003, 5.9 | 0.102, 205 | −0.52, 447 | −0.11, 1269 |
| Maricopa | 34 | −0.15 | 0.19 | −0.11 | 461 | 0.003, −6.0 | 0.122, 246 | −0.32, −65 | 0.15, 890 |

From Tables 7 and 8, note that all mean values for the period 1979–2020 were considerably less than the dimming period mean. For example, the gridMET mean was 235.9 W/m² and GEBA corrected was 245.4 W/m². No mean of any combination of datasets of the period 1979–2020 exceeded the dimming period mean of GEBA. All confidence interval data support the notion that the dimming period mean exceeds the so-called brightening period. There were considerable differences among the series post-1979. As mentioned previously, the gridMET series is said to not include features such as air pollution and should be more representative of regional change. As a consequence, there is a significant brightening effect detected with the regression data. The 41 year time series of gridMET for Phoenix showed a significantly increasing trend (r = 0.70, p = 0.000) with small bias and standard errors and adequate 95% CIs. The shorter time frame of 34 years (gridMET[b]) also showed similar metrics. Using the regression relation in Table 8, we obtained for Phoenix a value of +9 W/m² (+7 W/m²) for an estimate of brightening during the 1979–2020 (1987–2020) periods. This amounts to a +4.6% (+3%) increase for the two respective periods. Statistics obtained from [15] for gridded data of gridMET for 1979–2020 are +5.2 W/m² (Southwest region), and for point locations of gridMET, +2.4 W/m² (Las Vegas), +7.6 W/m² (Aguila), +10 W/m² (Encanto), +8.8 W/m² (Maricopa), +8 W/m² (Tucson), and +3.6 W/m² (El Centro). All trends were positive and ranged from +2.4 W/m² to +10 W/m² for the

region. It is not evident that urban grid points have necessarily slowed in brightening from these data.

Noting the results of Tables 7 and 8 for the surface sites, there was a significant downward trend for Encanto, albeit with lower confidence (r = −0.44, A = −0.29, B = 816) with a large range of confidence intervals for A and B coefficients. The downward trend is most likely because Encanto is an urban site exposed to significant particulate pollution (as illustrated in the next section). As noted by many authors, some sites in industrialized areas have shown reductions or leveling off since the 1980s [24–27]. The surface stations of Aguila and Maricopa show no statistically significant increases post-1987, even though they are considerable distances from the immediate urbanized area of Phoenix. However, this central Arizona desert region is one of natural aerosol production as well as urban effects. What likely accounts for the lack of brightening for rural sites may be in the findings of [28]. In that study, it was pointed out that the 1988–2009 trends in aerosols in the western United States were less negative than other parts of the country, and in some cases, the trends were even positive. The efficacy of an urban effect at Encanto is investigated in Section 3.3.

### 3.3. Rural–Urban Differences in Shortwave Irradiance

In the case of the Phoenix area, there have been several past investigations of urban effects on shortwave irradiance by urban climatologists. These have involved measurement for short time periods with automobile transects [29] and analyzing data in rural and urban settings with fixed sites [30]. This latter study included an analysis of many urban and rural AZMET sites. Thus, using an urban AZMET site compared to rural sites, we expected to find an urban effect on shortwave irradiance. From these past analyses, we know that the choice of Maricopa is a good representative of a rural site [30]. We used one year as an example to compare an urban site (Encanto) and one outside of the city (Maricopa). Figure 5a illustrates the seasonal progression of clear sky solar noon transmissivity (which we label T), derived from ground level shortwave irradiance as a fraction of extraterrestrial radiation for the latitude/longitude and time of year and hour for each site and day [23] in this plot for the Maricopa and Encanto stations.

Clear sky T ranges from 0.68 (Encanto) to 0.77 (Maricopa) during the year with considerably lower values at the urban site of Encanto compared to the rural site Maricopa, notably in winter and fall. We assumed that visibility reported at Sky Harbor airport might relate to these data, but found that for all but two days, visibility for all daylight hours was reported as 16 kms. The only incidence of lower visibility on any of the days was in the early morning hours for a few sample days. We hope to acquire other measures of visibility for further analysis.

Overall, summer decreases in T at both sites are controlled by ozone and water vapor of the summer months; the latter associated with monsoon effect intrusions of moisture [31,32], especially at the end of June to mid-September. At each AZMET site, irrigation is maintained, especially through hot summer months, and humidity differences were actually slightly higher at the rural agricultural site than at the urban golf course site [11]. However, evident in fall and winter were large differences in T values that reflected higher PM10 in the urban area than in rural locales (as seen in the Figure 5b plot of mean urban versus rural differences in PM10 from the sites shown in Figure 1). Percent differences in T exceeded 10% during winter. During summer, T differences dropped to half these values as PM10 in rural areas typically climb to approaching urban levels [13]. During 2019 at Phoenix in November and December, there was considerable rainfall (i.e., 57 mm), which had an impact on raising humidity and lowering T for clear days during these two months (after day 300). Typically, larger differences of PM10 would occur between rural and urban sites as the fall–winter pollution season ensues and mixing heights were lowered. The seasonal patterns of T were similar to [18]'s analysis of T for the years 1968–69. The annual mean difference of T was 3%.

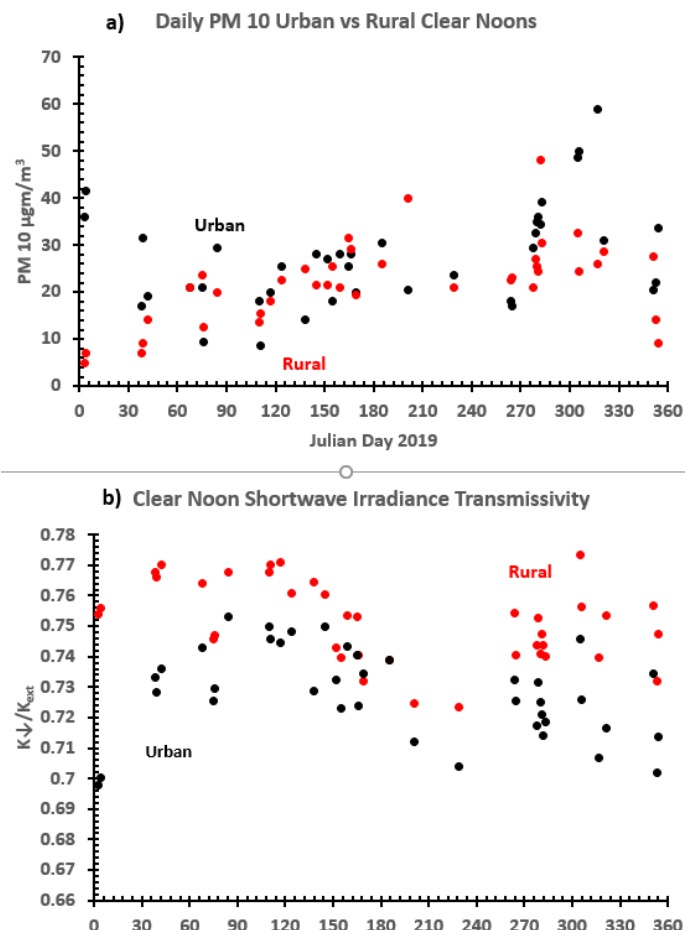

**Figure 5.** (**a**) Transmissivity T (K↓/K$_{etr}$) for urban site Encanto (black dots) vs. rural site Maricopa (red dots), (**b**) Urban vs. rural daily PM10 values. Urban (black dots) is mean of West Phoenix and Durango Complex; rural (red dots) is mean of Alamo Lake and Sacaton. Since data were for selected separate days over time, no curves were fit to the data.

We determined the correlation of Maricopa–Encanto differences in transmissivity (from Figure 5b) versus differences in urban minus rural PM10 (from Figure 5a). With N = 39, the r value was 0.55, *p* = 0.000, indicating larger differences in T for higher PM10 in the urban environment than in the rural surroundings. Our r value was similar to those reported by [33,34], who compared PM10 to urban/rural global radiation attenuation in % on a daily basis.

## 4. Conclusions

In our study, the work of [18] for the 1950–1970 period and research of [19] help to shed light on, and validate the GEBA solar records for the urban Phoenix airport station. The records show dimming observed in central Arizona over this time period and for our dimming analysis period of 1950–78. The estimate of the dimming for 1950–70 for the Phoenix site is a reduction of 25 W/m$^2$ (−9.5%) for all sky conditions and 20 W/m$^2$ (−7.0%) for clear days. For the longer dimming period of 1950–78, the all sky condition dimming was 30 W/m$^2$ (−13%). These are especially high values of dimming relative to other reported values in the literature [1,7]. We think that this is due to a combination of air pollution and desert aerosols in the Salt River Valley of a rapidly growing and large city in a desert climate, with persistent winter and fall inversions, low winds, and many clear days per year. The exact causes of these changes remain for further research, especially the role of cloudiness trends and remaining uncertainty in the magnitude of dimming due to instrument changes and calibration history, especially prior to 1966, a time when the

National Weather Service expanded their efforts at calibrating equipment and planning for better sensors across the country.

For the post-1978 period, and using our statistically significant regression relationships, the gridMET time series of 1979–2020 showed a brightening effect of 9 W/m$^2$ (+4.6%); for the 1987–2020 (during the period of the AZMET network), the values were 7 W/m$^2$ (+3.0%). This is in line with the U.S. values reported in [1,2]. The only significant change among the three AZMET sites analyzed for the 1987–2020 period is for the urban site of Encanto with a change of −9 W/m$^2$ (−3.8%). Aguila and Maricopa, although out of the immediate locale of the urban area, showed no statistically significant brightening or dimming from 1987–2020. The reduction of −3.8% at Encanto with no change at Maricopa matches well with our analysis of the rural–urban transmissivity difference for 2019 of 3%.

For southern Arizona, [28] found decreasing trends in most aerosol chemical constituents except dust. It was shown in [28] that there is seasonality in the type of aerosols influencing the region. March to July experiences mineral dust; from May to August, large wildfire activity and organic aerosols also contribute. Positive trends in aerosols were thought to be due to a dust influence. Trend analyses done by [28] for 1988 and 2009 indicate that the strongest statistically significant trends were reductions in sulfate, elemental carbon, and organic carbon, and increased in fine soil during the spring (March–May) at select sites including Phoenix. Regional PM10 reached the highest levels in the summertime (May–August) except for Phoenix. The different monthly behavior of PM10 in Phoenix resulted from anthropogenic activity such as vehicles and fugitive and wind-blown dust from agricultural fields, roads, and construction activity [28]. The monthly trend of aerosols showed a pronounced peak between April and July, when meteorological conditions promote dust emissions. The general monthly trends suggest that dust aerosol is a significant contributor to PM10 in the region [28].

The implications of the findings of dimming and brightening are many and have been discussed by [1,2,35]. We have not explored other databases that have been produced for applications of energy assessments [36]. We intend to explore further analysis of these data and models relative to our findings. There appears to be increasing opportunities to conduct retrospective and ongoing analysis of solar records that are part of new weather networks in and around urban areas [9]. These data can show the variability of the solar resource across rural and urban areas and are starting to represent longer term databases to potentially be part of a global dimming/brightening analysis. Desert environments experience many days of clear skies year round, calm winds, and local inversions plus pronounced seasonal variations of mixing heights. Urban effects and long-term trends perhaps ought to be more apparent in such environments with considerably less cloud cover and abundant numbers of clear days. In addition to urban and rural variability shown by many researchers, it appears increasingly possible to link local variability detected in special weather network data to long-term trends caused by global scale processes.

**Author Contributions:** Conceptualization, A.B. and R.T.; Methodology, A.B.; Software, A.B.; Validation, A.B. and R.T.; Formal analysis, A.B.; Investigation, A.B. and R.T.; Resources, A.B.; Writing—original draft preparation, A.B.; Writing—review and editing, A.B. and R.T.; Visualization, A.B. All authors have read and agreed to the published version of the manuscript.

**Funding:** This research received no external funding.

**Institutional Review Board Statement:** Not applicable.

**Informed Consent Statement:** Not applicable.

**Data Availability Statement:** Restrictions apply to the availability of GEBA data. Data was obtained from [https://geba.ethz.ch/ by permission of Martin Wild].

**Acknowledgments:** We are indebted to anonymous reviewers for enlightening us on the work of Martin Wild, who then graciously gave us permission to access the GEBA data. We thank Barbara Trapido-Lurie for all cartography. A.B. thanks R.T., who conducted field work in early 2000s on solar radiation in the Phoenix area. A.B. owes him a great debt of gratitude for ideas and working with him on this paper. We thank our colleague Ariane Middel, Professor at ASU, who archived much of the data from AZMET for analysis; Ron Pope of the Maricopa County Air Quality Department for suggesting access to PM10 daily records for Arizona PM10 monitoring stations; and Nancy Selover, former state climatologist of AZ, for the metadata of the sites. We further appreciate editing of Mary Hoadley.

**Conflicts of Interest:** The authors declare no conflict of interest.

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
