# Peer review of "Shortwave Irradiance (1950 to 2020): Dimming, Brightening, and Urban Effects in Central Arizona?"

_climate, doi:10.3390/cli9090137_

Round 1
Reviewer 1 Report
I reviewed the re-submitted paper “Monthly Shortwave Irradiance (1950 to 2020): Dimming, Brightening, and Urban Effects in Central Arizona?”. But from what I can see author did not followed recommendations provided in first review. For example, comment number 15 is about to expand introduction because it is pretty short; and the answer received is just “redone”, but it remains too short. Also comment number three is about highlighting objective in abstract, the answer was “Onjectives redone” and objective does not appear in this section. Please check the list of comments I shared in the first round.
In my opinion, before the paper is accepted, author should prepare a revised version in which all comments of the first reviewed must be addressed.

Author Response
attached is response

Reviewer 2 Report
The paper has improved but still needs revision. The introduction is short, I would expect to find something about dimming and brightening trends from previous studies. The methodology section is unclear and missing information which makes it hard to read the result section. In the methodology I would expect to read details about the which stations instrumentation and positioning (urban/rural), clear sky selection and calculations of the transimissivity. The result sections fitting routines look ok but are missing uncertainty intervals which can for example be calculated using repeated bootstrapping or cross-validation. Please include the 95% range of the fits. A bit more statistics like RMSE values would be nice. Would it be possible to include a table with stations dimming and brightening trends for different periods? This would help the discussion and conclusion section. Line 21. What do the authors mean with “mixed” and “thereafter”? Is there a pattern in the whole of Arizona? Are the dimming and brightening trends derived from the annual time-series or is this a long term trend? Line 25. Please remove the abbreviation AZ from the keywords. Line 91. I do not understand how the change in clear days can be derived from Figure 2. This is only a comparison between the two datasets. From table 1 I suspect the comparison is only for 1 station? What is the uncertainty of the fit? What is the cause for the 3.6W/m2 offset? Line 97. Are 270W/m2 and 235W/m2 the values for 1950 and 1970? How was the -13% calculated? Shouldn’t this be part of the result section? Line 99. Why must the decreasing irradiance be subscribed to calibration issues? Line 174. What kind of clear sky data was used? How is this measured, what were the selection criteria for a clear sky day? For which area was the station representative. Please elaborate. Line 218. What do the black and red line represent? Line 231. Are a total of 2 stations per location (urban/rural) sufficient to draw conclusions from? Is there a difference between the two rural sites, since one of them is located closer to the city? Line 279. In the beginning of the report a reduction of 10W/m2 was reported. Which 25 years? Line 282. Shouldn’t this be part of an introduction?Author Response
see file

Reviewer 3 Report
The authors have addressed most comments in the previous review. The manuscript is obviously improved.
Minor comments
#1 The words, Dimming, Brightening, and shortwave irradiance sound confusing particularly for non-expert readers interested in this work. I would suggest authors chose more common words (for example, if Brightening is the opposite of Dimming use only one word. Is shortwave irradiance same as transmissivity? I suggest authors discuss if visibility can be inferred from Dimming, Brightening (L299-302).)
#2 L23: ".. large reductions .. to 2020" is not clear. Large reduction in what?
#3 L277-278: This is not clear. Does it mean any trends are from natural causes.
#4 Abstract and Conclusion sections: What are the inplications of the findings of this study? Why the increasing/or decreasing brightening/or dimming matter (for environment, ecology, health etc)?
Author Response
attached is response

Reviewer 4 Report
Manuscript revision: Climate ID-1330996 2nd August 2021
Title: Monthly Shortwave Irradiance (1950 to 2020): Dimming, Brightening, and Urban Effects in Central Arizona?
Authors: Anthony Brazel1 and Roger Tomalty
School of Geographical Sciences & Urban Planning and Urban Climate Research Center, Arizona State
University, Tempe, AZ 85287-5302, USA; abrazel@asu.edu. 10301 N. 70th St. #12, Scottsdale, AZ 85253; rtomalty@gmail.com.
Abstract: Three Agricultural Meteorological Network (AZMET) weather stations in central Arizona have observed shortwave irradiance over the period 1987-present. Monthly data were compared to gridMET data, 4-km gridded climate data based on NLDAS-2. The gridMET data spans the period 1979 to 2020. Two of the more rural AZMET sites used are located north and south of the Phoenix Metropolitan Area, and another site is in the center of the City of Phoenix. AZMET data were compared to gridMET for the overlapping period 1987-2020. Global Energy Budget Archives (GEBA) were accessed for the available years 1950-1994 for Phoenix, Arizona and compared to gridMET data for the overlapping period 1979 to 1994. The analysis of AZMET vs gridMET and gridMET vs GEBA overlapping periods resulted in highly significant correlations.. Therefore, a time series of shortwave irradiance data for 1950-2020 was constructed for monthly and annual time scales for central Arizona. Using a combination of GEBA and gridMET data, the resultant monthly and annual time series demonstrate dimming up to early 1980s, with a mixed signal of brightening and dimming thereafter. Using these data prior to 1987 linked with AZMET data illustrates a brightening at rural sites up to mid-21st century. The urban site experienced little brightening post mid-1980s and large reductions to 2020. All AZMET sites including the gridMET data display a tendency for reductions of shortwave irradiance during the last decade.
Keywords: shortwave irradiance; Phoenix, AZ; GEBA; gridMET; AZMET; dimming and brightening; trends; urbanization
Reviewer Recommendations:
1-The novelty of the study should be mentioned or explained at the end of the Introduction.
2- If the manuscript topic is studied for the first time in the geographical area, the interest and importance of the subject should be emphasized.
3- The different sections that compose the manuscript should be shown briefly but with enough detailed at the end of the Introduction section.
3.- In this type of manuscripts is very interesting de number of data used in each station for the corresponding evaluation. For this information, a new Table with the type and number of data in “each calculation” must be included with clarity and right written. With this objective, Section 2 should be composed by these three subsections: Site, Measurements or Data used and Methodology. About data used, the subsection DATA and the new Table will compose by the following information for each place: In columns: Place name, N (number of data), Initial date of data, Final date of data, maximum value, and minimum value of the variable; in total 6 columns and the number of lines depend on the station number . All variables with units. As consequence, people that read manuscript will know the basic data characteristic
4- Data Quality Control should be evaluated, and it should taken into account and indicate briefly also at the end of the Introduction, if it is possible.
The following references are recommended for this activity:
DOI: 10.1016/j.atmosres.2014.02.007
DOI: 10.1029/2011JD015836
Quality control data is interesting for selecting the best data of the series and the defective data will be eliminated. Due to the interest of the data methods the reference recommended should include in the manuscript references list.
5-Figure 1 should include the latitude and longitude over axis Ox and OY. In order to have a more real situation of the measurement stations.
6-Figure 1: the decimal values better with 2 decimal figures. If the value that is eliminated is bigger than 5, the previous values should be increase in one unit. Example: 4.0789 will be: 4.08 .
7-Figure 2: Equation of the line over the figure: as before, better the decimal figures only with two decimal numbers. The number of decimal figures should correct along the manuscript and on the Figures and Tables.
8-Please the scale visions in Figures 2 and 3 should be similar. Manuscript shows different graduation scales. Along the manuscript the Figures should show similar “format”. For example Figures 2 and 3, looking at OY axis: better with tick marks (for clarify normally an axis has short marks and some longer marks). It is said, some thick marks should be bigger than others. Please take care with tick marks and tick labels, and try to write similar format in all Figures along the manuscript.
The same that before (number 8) about the decimal number figures that appear in the plane over Figure 2 and 3.
9-I do not see if “W is in capital letter over the Figures. Please verify.
10-Figures 4 and 5 are trend values or monthly values each year, in Figure 4 ???
11- Figures 4 and 5 , the axis OY have all thick marks in same sizes and this complicate the value observations. There is not units on the axis. Complete the Figures with all previous details recommended in point 10 of this revision. Thanks
12- The values of R2 in all Figures, better with only two decimal numbers. For example R2=0.67
13-Line 53- This line should be “Place, Data and Methodology”. In addition, it should be divided in subsections with order and clarity. Please include also the data quality control at the data section.
14-Line 65: The word “desne” is correct…”???
15-Line 272: Normally “Results and Discussion” should be one section or two as manuscript prefers; “Conclusions” another section and “References” the last section. But here, it is clear that Conclusions is one independent section. Please try to obtain a “clear” “Conclusions” and write it as a section.
16-Lines 317-323: Due to the manuscript has more than one author, in this paragraph and when referring to the authors, in my opinion, it should write "we" and not "I" or “the authors”.
Author Response
see file

Round 2
Reviewer 2 Report
The manuscript has improved significantly and the authors have revised the manuscript properly. Only minor spell checking required before publication.